# Energy Metabolism Changes and Dysregulated Lipid Metabolism in Postmenopausal Women

**DOI:** 10.3390/nu13124556

**Published:** 2021-12-20

**Authors:** Seong-Hee Ko, YunJae Jung

**Affiliations:** 1Department of Microbiology, College of Medicine, Gachon University, Incheon 21999, Korea; marialovegod625@gmail.com; 2Lee Gil Ya Cancer and Diabetes Institute, Gachon University, Incheon 21999, Korea

**Keywords:** estrogen deficiency, energy metabolism, changed body composition, metabolic rate, lipid metabolic disorder

## Abstract

Aging women experience hormonal changes, such as decreased estrogen and increased circulating androgen, due to natural or surgical menopause. These hormonal changes make postmenopausal women vulnerable to body composition changes, muscle loss, and abdominal obesity; with a sedentary lifestyle, these changes affect overall energy expenditure and basal metabolic rate. In addition, fat redistribution due to hormonal changes leads to changes in body shape. In particular, increased bone marrow-derived adipocytes due to estrogen loss contribute to increased visceral fat in postmenopausal women. Enhanced visceral fat lipolysis by adipose tissue lipoprotein lipase triggers the production of excessive free fatty acids, causing insulin resistance and metabolic diseases. Because genes involved in β-oxidation are downregulated by estradiol loss, excess free fatty acids produced by lipolysis of visceral fat cannot be used appropriately as an energy source through β-oxidation. Moreover, aged women show increased adipogenesis due to upregulated expression of genes related to fat accumulation. As a result, the catabolism of ATP production associated with β-oxidation decreases, and metabolism associated with lipid synthesis increases. This review describes the changes in energy metabolism and lipid metabolic abnormalities that are the background of weight gain in postmenopausal women.

## 1. Introduction

Women generally experience natural menopause due to loss of ovarian follicle activity between the ages of 45 and 55, and surgical menopause is also accompanied by loss of estradiol (E2) [1,2]. In general, women live longer than men, and life expectancy is increasing worldwide. The median age of women will reach 82 in developed countries by 2025 [3]. Thus, about half of a woman’s life is after menopause. It is well known that hormonal changes are one of the significant physiological effects of menopause. There are three types of estrogen—estrone (E1), E2, and estriol (E3)—all of which are C18 steroids and aromatic molecules [4]. Each form has a unique function that matches a woman’s life cycle characteristics, such as reproductive age, pregnancy, and menopause [5]. Before menopause and throughout the menstrual cycle, meningeal cells produce androstenedione, which acts as a metabolic precursor to E1 and testosterone in the ovaries and peripheral tissues [6]. In granulosa cells, androstenedione is converted to E1 by aromatase activity of CYP 19, after which E3 is converted to E2, with an average level of total estrogen of 100-250 pg/mL [6] (Figure 1). By contrast, circulating E2 levels drop sharply to 10 pg/mL in postmenopausal women [7], meaning that women spend half of their lives in estrogen deficiency.

With these hormonal changes, menopausal women are more likely to experience various metabolic disorders such as dysregulated lipid metabolism, fat redistribution, visceral fat accumulation, and altered fatty acid metabolism [8,9,10]. Furthermore, they are readily accompanied by changes in body composition and energy metabolism, loss of muscle volume and strength, and weight gain [11,12,13].

To better understand metabolic alterations, it is important to understand recommended body weight, body composition, and the basic concepts of energy metabolism, including basal metabolism, resting energy expenditure (REE), and metabolic rate. This review describes the concepts needed to understand energy metabolism, including the composition of the human body and the components of energy expenditure, and discusses possible changes in energy metabolism in postmenopausal women. By summarizing overall energy metabolism, this review provides information for preventing muscle and bone loss. In addition, it also addresses lipid metabolic changes related to visceral fat accumulation, lipolysis, fatty acid oxidation, fat redistribution, and weight gain.

## 2. Menopause Associated Changes in Energy Metabolism

### 2.1. Composition of the Human Body

The chemical compositions of the male and female human bodies are shown in Table 1, based on average physical dimensions from the measurements of thousands of subjects who participated in various anthropometric and nutrition surveys [14]. As seen in Table 1, the muscles of the reference man account for 44.8% of body weight, compared to 36% for women. The reference man has 15% total body fat versus the female’s 27%. The reference woman’s storage fat accounts for 15% of her weight, and essential fat accounts for 12%. The total amount of our body fat is composed of storage and essential fats. The latter is present in lipid-rich tissues throughout the central nervous system, bone marrow, heart, lungs, liver, kidneys, spleen, intestines, and muscles [15]. In females, essential fat also includes fat in mammary glands and the pelvic region. Fat that accumulates in adipose tissue is called storage fat, and triglyceride (triacylglycerol) is the typical component, which accounts for ~86% of body fat [16]. Triglyceride can be broken down into fatty acids and glycerol by lipase and used to produce ATP for energy metabolism [13]. In other words, triglyceride can be used as an energy source for heat generation in the human body. Fat mass is, therefore, the most changeable constituent of the body. In terms of muscle and body fat mass, the body compositions of men and women are distinctly different, with women having more fat and less muscle mass [17].

#### Changes in Body Composition in Postmenopausal Women

As women age, they tend to have an increased amount of fat tissue in the abdomen and relatively reduced fat in the hip-thigh area. Changes in body composition are also involved, including reductions of fat-free mass (FFM) and lean body mass (LBM) [18]. Women who experience natural menopause have changes in body fat mass, such as a decrease in total leg fat and an increase in abdominal fat [19]. It is likely that these changes are at least in part due to hormonal changes that occur when women have high levels of androgens versus E2 after menopause [20]. Several researchers have shown that menopause itself influences upper body fat distribution independent of aging [21,22,23]. However, many cross-sectional studies using dual-energy X-ray absorptiometry showed that postmenopausal women have lower FFM or LBM in the whole body, trunk, and lower extremities than premenopausal women [24]. In a 6-year follow-up longitudinal study, natural postmenopausal women lost more FFM than age-matched premenopausal women while showing increased central adiposity and reduced energy expenditure during rest and physical activity [25]. Interestingly, one out of five relatively healthy Korean postmenopausal women aged over 65 years exhibited a decline in muscle mass, and 7.6% of subjects showed declines in both muscle mass and strength [26]. The study also reported the intensified loss of skeletal muscle with aging [27].

### 2.2. Component of Energy Expenditure

Menopausal women are likely to experience changes in lipid metabolism along with weight gain [8]. To understand why this occurs, it is necessary to know the basic concepts of energy expenditure that influence body weight and composition, the function of the basal metabolic rate (BMR), the thermic effect of food, and physical activity. Total energy expenditure consists of (1) the BMR or REE, (2) the thermic effect of food (TEF), and (3) the thermic effect of activity (TEA) (Figure 2).

#### 2.2.1. BMR

Basal metabolism is the energy expended by internal processes during a period of complete rest in a climate-controlled environment at least 10–12 h after consumption of the most recent meal; that is, the minimum amount of energy needed to sustain life processes [28]. Typically, 50–65% of total energy expenditure is attributable to the BMR that is commonly used with REE, which is simply the basal metabolism during a non-active state in a climate-controlled environment at least 10 to 12 h after the consumption of the most recent meal (Figure 2) [29]. However, the most important difference between the BMR and REE is that a subject does not need to fast to measure REE.

#### 2.2.2. TEF

The TEF refers to the metabolic reaction of food (i.e., heat generation) due to its digestion, absorption, processing, and storage [30]. TEF can increase metabolism by more than 5–15% compared to the BMR when consumed in large portions rather than small frequent meals, when carbohydrates and protein rather than dietary fat are consumed, and when a low-fat plant-based diet is followed [31]. Generally, TEF is estimated as 10% of total energy intake during a particular period (Figure 2) [32]. For example, TEF can be estimated at 160 kilocalories for an individual who consumes a mixed diet containing 1600 kilocalories over a 24-h period.

#### 2.2.3. TEA

The TEA refers to skeletal muscle activity associated with the maintenance of position and posture, as well as skeletal activity during obvious movements such as walking, running, swimming, climbing stairs, or vacuuming [28]. Thus, TEA is a highly changeable component. While the contribution of skeletal muscle activity may seem trivial, simply sitting on a chair without back support augments heat generation by 3–5% [28]. This increase in metabolism is much more significant when standing [33]. Average physical activity accounts for 20–40% of total energy expenditure, but this depends on the individual’s physical activity (Figure 2) [34]. Therefore, sedentary people who are less active consume much less energy and may have lower energy than more physically active people [35]. These results indicate that a sedentary lifestyle before menopause may contribute to a decrease in total energy expenditure [36].

### 2.3. Metabolic Rate Difference between Skeletal Muscle and Adipose Tissue

The human body consists of protein, water, minerals, and fat, which can be largely divided into fat mass and FFM [37]. FFM is often conflated with LBM, which is calculated by subtracting fat mass weight from total body weight. LBM is the weight of internal organs, skin, bones, body water, tendons, and muscle mass [38]. The human body is composed of various tissues and organs, each of which has a specific function and mass, resulting in a different contribution to the BMR. FFM or skeletal muscle accounts for approximately 60–85% of the body mass; it is regarded as an energy consumer and a major determinant of the BMR or REE. Therefore, the metabolic energy rate differs depending on the body composition ratio [34]. In this regard, it is necessary to consider the difference between the energy consumptions of skeletal muscle and adipose tissue, which are 13 kcal/kg/day and 4.5 kcal/kg/day, respectively (Table 2) [39]. The energy expenditure of skeletal muscle is three times higher than that of adipose tissue, so the BMR is higher for individuals with high muscle mass, even if they have the same weight as a person with low muscle mass. During aging, fat increases at the expense of FFM; moreover, the loss of LBM due to sarcopenia (a decrease in skeletal muscle) and an increase in adipose tissue will result in a lower BMR [40].

#### 2.3.1. Changes in Body Composition and Energy Expenditure in Postmenopausal Women

As aging progresses, FFM or skeletal muscle loss occurs, which leads to a decrease in the BMR [41,42]. Considering that the metabolic rate of skeletal muscle is more than three times higher than that of adipose tissue, loss of skeletal muscle or FFM may lead to a decrease in BMR among menopausal period women [43,44]. The decrease in E2, along with changes in body composition in postmenopausal women, plays an important role in regulating adipocyte differentiation and distribution. Gavin et al. reported that ovarian hormone depletion after ovariectomy (OVX) increased the production of bone marrow-derived adipocytes (BMDAs) in mice visceral fat depots that are gonad adipose depots [45]. In addition, E2 replacement continued to dampen the accelerated production of BMDAs. Furthermore, estrogen receptor (ER)α genetic knockdown enhanced BMDA production in both the gonadal and inguinal depots, which demonstrated that E2 regulates BMDA production [45]. If this is translatable across species, it suggests that the production of BMDAs occurs through a mechanism where visceral fat increases in estrogen-deficient postmenopausal women.

Estrogen plays a pivotal role in systemic energy homeostasis. OVX mice have been shown to reduce systemic O_2_ consumption and energy expenditure, leading to the weight gain associated with increased body fat. On the other hand, exogenous E2 supplementation increased systemic O_2_ consumption and energy expenditure, resulting in increased systemic insulin sensitivity in OVX mice [45]. This provides a mechanism by which E2 supplementation may ameliorate insulin resistance [45]. With regard to energy homeostasis, it was also reported that E2 directly regulates mitochondria membrane biophysical properties and bioenergetic functions, providing a direct pathway by which E2 states broadly influence energy homeostasis [46,47]. OVX mice exhibit decreased mitochondrial respiratory function, cellular redox state, and insulin sensitivity in skeletal muscle [46]. E2 may be a mitochondrial membrane component in many tissues that locally affects bioenergetic activity and energy homeostasis. Additionally, Gavin et al. showed that reducing endogenous E2 in humans reduced energy expenditure and increased visceral fat [45]. It was also reported that estrogen deficiency increased BMDA accumulation in the white adipose tissue of mice and was associated with decreased physical activity in women [48].

Postmenopausal women may experience body shape changes due to increased loss of FFM or LBM with high energy metabolisms, while fat tissue increases with low energy metabolism and increased BDMAs [45]. Because the E2 state has a widespread impact on energy homeostasis, a decrease in levels of this hormone reduces systemic O_2_ consumption and energy consumption. Postmenopausal women’s bodies also consume less energy to maintain their basic life processes [49]. Collectively, these changes may affect the BMR and lead to insulin resistance along with weight gain.

#### 2.3.2. Sarcopenia in Postmenopausal Women

In general, skeletal muscle mass and muscle strength peak in the mid-20s and 30s and then gradually decrease [50]. Sarcopenia refers to the degenerative loss of skeletal muscle that occurs at a rate of 3–8% every 10 years after the age of 30 years and accelerates with age [8]. This condition is associated with increased risks of functional disability, falls, fractures, and overall mortality among the elderly [51]. Women develop sarcopenia earlier than men, and the decline of skeletal muscle mass and strength accelerates with the onset of menopause [52,53]. When sarcopenia coexists with osteoporosis, it results in a geriatric syndrome called “osteosarcopenia,” which increases the risk of weakness, hospitalization, and death [54]. In a prospective cross-sectional study by Buliana et al., the prevalence of osteosarcopenia was reported to be high among postmenopausal women with an increased risk of fracture [55]. Patients in the osteosarcopenia group had a greater risk of frailty than patients in the osteoporosis-alone group (odds ratio, 2.33; 95% confidence interval, 1.13–4.80; and *p* = 0.028). According to a retrospective observational study, one-year mortality of osteosarcopenia (15.1%) was higher than that of other groups (normal, 7.8%; osteoporosis only, 5.1%; and sarcopenia only, 10.3%) [56]. In postmenopausal women, insufficient protein and calcium intake and low levels of physical activity appear to be the most common risk factors for osteosarcopenia [55]. Generally, it is accepted that menopause is associated with accelerated loss of FFM or skeletal muscle, which further decreases energy expenditure during rest and physical activity [57]. Thus, body function impairment originates from muscle loss in menopausal women, leading to difficulties in carrying out voluntary activities and reducing the quality of life. The most promising strategy for increasing muscle and bone mass is resistance training, as well as sufficient amounts of protein, vitamin D, calcium, and creatine to help preserve these tissues during menopause [53].

## 3. Background of Weight Gain in Postmenopausal Women

### 3.1. Lipid Metabolic Abnormality Due to E2 Hormonal Change

Typically, women spontaneously experience menopause between the ages of 45 and 55 because of decreased ovarian follicular activity [1]. Menopause does not happen all at once; rather, it progresses through a transition period to the postmenopausal stage. With regard to surgical menopause, bilateral oophorectomy has been shown to cause dyslipidemia and significant loss of bone density within one year [2]. Around this time, the female hormones undergo drastic alterations. One of the major physiological changes associated with menopause is a sharp decrease in E2 that contributes to lipid metabolic disorders. Notably, women tend to develop more cardiovascular disease after menopause due to estrogen deficiency and alterations in lipid metabolism [58]. The reason is due to the unique role of E2 that is synthesized using low-density lipoprotein cholesterol (LDL-C) in the ovary (Figure 1). Therefore, a decrease in E2 synthesis due to menopause means that LDL-C is no longer used for synthesizing E2, so it remains in the systemic circulation (Figure 1). Since postmenopausal women have high LDL-C levels, there is an increased risk of metabolic syndrome symptoms, including central obesity, insulin resistance, dyslipidemia, hypertension, and cardiovascular disease [59].

### 3.2. Fat Redistribution in Postmenopausal Women

Most women tend to experience changes in the composition of the body as they get older, and this period almost perfectly coincides with menopause [60]. In premenopausal women, adipose tissue is predominantly distributed in the gluteal femoral subcutaneous compartment, whereas postmenopausal women tend to exhibit higher total body fat mass, fat percentage, and accumulation of central fat [61]. This can be partly explained by the changes in circulating endogenous sex hormone levels, because estrogen androgen receptors are expressed in both visceral and subcutaneous adipocytes [62,63]. Estrogens bind to ERα and ERβ, and androgens bind to the androgen receptor, enabling sex hormones act on their target cells [64]. Therefore, the reduction of the circulating sex hormones will change their action in target cells.

Ovarian estrogens induce peripheral fat storage mainly in the gluteal and femoral subcutaneous regions that express ERα, which mediates lipoprotein lipase activity and triacylglycerol accumulation in adipocytes [65]. On the other hand, androgens—Primarily bioavailable testosterones—Augment visceral abdominal fat accumulation [63]. In postmenopausal women, the concentration of E2 in circulation decreases, so the androgen to estrogen ratio increases [8]. Therefore, relative androgen excess (a higher baseline testosterone/E2 ratio) causes weight gain and body fat redistribution in postmenopausal women [63,66]. According to the longitudinal, community-based, 5-year follow-up Study of Women’s Health Across the Nation, postmenopausal women showed twice as much visceral abdominal fat and subcutaneous adipose tissue than premenopausal women [67]. However, testosterone levels were similar among pre-and postmenopausal women [68], which suggested that fat redistribution may be affected by a marked decrease in estrogen levels as opposed to testosterone levels. Moreover, it was also observed that the androgen to estrogen ratio was also elevated in premenopausal women with polycystic ovarian syndrome [69,70].

### 3.3. Excessive Visceral Abdominal Fat and Metabolic Alterations

The basic role of adipocytes in lipid metabolism is storing energy in adipose tissue in the form of triacylglycerol and releasing it as free fatty acids (FFAs) as needed to provide fuel for working muscles [71]. Adipocytes also control glucose homeostasis by secreting glycerol and fatty acids, which play critical roles in hepatic and peripheral glucose homeostasis by mediating the breakdown of triacylglycerol [72,73]. However, as women experience menopause, dysregulated adipocyte metabolism occurs due to estrogen reduction, and various metabolic diseases appear [8].

Visceral adipose tissue in both humans and rats is composed of mesenteric, retroperitoneal, osteoporosis, and reproductive gland deposits, so rodent models have been widely used to simulate fat metabolism in the human body [74]. In animal models such as OVX and ERα knockout mice, loss of ovarian hormones has been demonstrated to increase BMDA production in the visceral fat depot and gonadal adipose depot [45]. ERα plays an important role in regulating the de novo synthesis of BMDA, which supports the observation of increased visceral fat in estrogen-deficient postmenopausal women [45]. In this setting, visceral fat increases, and abdominal obesity intensifies in obese menopause women. Since the rate of lipolysis differs throughout the body, abdominal fat has a higher rate of lipolysis than gluteal fat due to the higher rate of catecholamine-mediated lipolysis in the abdomen [66,75].

### 3.4. Alterations in Fatty Acid Metabolism

It is widely known that FFAs are produced by the excessive decomposition of visceral fat, which promotes hepatic insulin resistance in connection with the increased flow of FFAs to the liver [76]. Interestingly, in a metabolomic study comparing the evaluation of fatty acid metabolism between pre-and postmenopausal women, fatty acid metabolites such as heptanoate, octanoate, and pelargonate were significantly higher in the visceral fat but not in the subcutaneous fat of postmenopausal women. This means that increased lipolysis of visceral fat may trigger an accumulation of fatty acid metabolites [77].

Extensive studies have been conducted on the role of estrogen in various metabolisms, immunity, and inflammatory processes in rodents and humans [78]. Thanks to more sensitive and accurate proteomic techniques, a number of proteins and pathways in visceral fat were found in OVX rodent models [79]. Interestingly, Boldarine et al. reported that OVX induced the upregulation of genes related to lipogenesis and downregulation of genes related to fatty acid oxidation [10]. It was reported that expression of the adipogenesis-associated gene all-trans-retinol 13,14-reductase (RETSAT) was increased in OVX mice. RETSAT is induced during adipocyte differentiation and is positively regulated by the transcription factor, peroxisome proliferator-activated receptor γ, which means that upregulated RETSAT indicates active fat accumulation in OVX mice [80]. Moreover, adipose tissue lipoprotein lipase (AT-LPL) showed increased expression, indicating a high ability of retroperitoneal adipose tissue to absorb lipoprotein-derived FFAs. AT-LPL is an enzyme that decomposes triglyceride into FFAs for absorption and storage by adipocytes, and plays an important role in fat accumulation and fat storage distribution. Therefore, it can be expected that increased uptake of lipoprotein-derived FFA in retroperitoneal adipose tissue due to enhanced expression of AT-LPL could be expected to affect the TG synthesis pool in OVX mice [72]. Estrogens are also associated with inflammatory responses in women. Although estrogens are known to enhance autoimmune diseases [81], decreased estrogen increases susceptibility to infectious diseases, as demonstrated by the defective innate immune responses against viral infection in OVX mice [82]. In addition, OVX enhanced the susceptibility of female rats to dyslipidemia with a decrease in innate cytokines, suggesting impaired metabolic and immune homeostatic responses with the loss of estrogens [83].

A pathway analysis study reported that OVX affected the fatty acid metabolism/mitochondrial fatty-acid-oxidation pathway and fatty acyl coenzyme A (CoA) biosynthesis pathway in visceral adipose tissue [10]. Fatty acid catabolism mainly occurs in the mitochondria [84]. Long-chain fatty acids with more than 14 carbons are converted into fatty acyl-CoA and pass through the mitochondrial membrane [85]. Fatty acid catabolism begins with acyl-CoA synthetase that adds CoA to fatty acids using ATP in the cytoplasm [85]. Once it has passed through the mitochondrial membrane, acyl-CoA begins the beta-oxidation process [85]. FFA activation occurs by bonding with CoA and is promoted by ligase enzymes, which are important steps in the oxidation and synthesis of triacylglycerol and other lipids [86]. Fatty acid oxidation is the largest contributor to ATP production, accounting for 40–60% [87]. Therefore, a decrease in fatty acid oxidation leads to the accumulation of lipids in a state that the body cannot efficiently burn as a fuel source, resulting in insulin resistance [10,88].

Taken together, the loss of estrogen and increase in circulating androgens in postmenopausal women induce changes in body fat distribution, leading to abdominal obesity. In addition, loss of estrogen increases BMDA production in mice visceral fat depots, which are gonad adipose depots [45]. As such, obese menopausal women have increased visceral fat and aggravated abdominal obesity. The rate of lipolysis is higher in abdominal fat than in gluteal fat [77], which triggers excess FFA production from excessive visceral fat breakdown, promotes insulin resistance, and leads to metabolic diseases [76]. In the OVX model, the expression of genes related to β-oxidation and lipogenesis is decreased and increased, respectively. Therefore, excessive fatty acids produced by the breakdown of visceral fat cannot be efficiently oxidized as a fuel source in the body through β-oxidation, which leads to fat accumulation. This can cause unfavorable changes in both fat metabolism and energy metabolism [10].

## 4. Conclusions

Both natural and surgical menopause is accompanied by changes in body composition due to loss of E2 secretion, and various changes can occur in energy and lipid metabolism. Postmenopausal sarcopenia and increased fat mass change the energy metabolic rate and affect the BMR. In addition, postmenopausal women are susceptible to obesity, and weight loss becomes more difficult. Moreover, fat redistribution caused by hormonal changes leads to changes in body shape. In particular, as the amount of visceral fat increases, FFAs also increase due to excessive fat decomposition, which can lead to insulin resistance and cause metabolic diseases. In addition, excessive FFAs produced by the lipolysis of visceral fat after menopause are not properly used as energy sources through β-oxidation, which is because genes related to β-oxidation are downregulated following the loss of E2. In this review, energy metabolism and lipid metabolic disorders related to menopause have been comprehensively summarized in relation to the basic concepts of body composition and energy expenditure. An understanding of overall energy metabolism, osteosarcopenia, and a recognition of the importance of preventing muscle loss will help to address individual health needs. This review also systematically organized the background of weight gain due to E2 loss and lipid metabolic abnormalities. However, we did not discuss physical activity and nutritional therapy, which are prominent factors influencing metabolic changes. The relation between menopausal status and overall energy and lipid metabolism, particularly fatty acid oxidation, must be clarified to prevent the growing problem of obesity-related disorders, including insulin resistance and metabolic syndrome.

## Figures and Tables

**Figure 1 nutrients-13-04556-f001:**
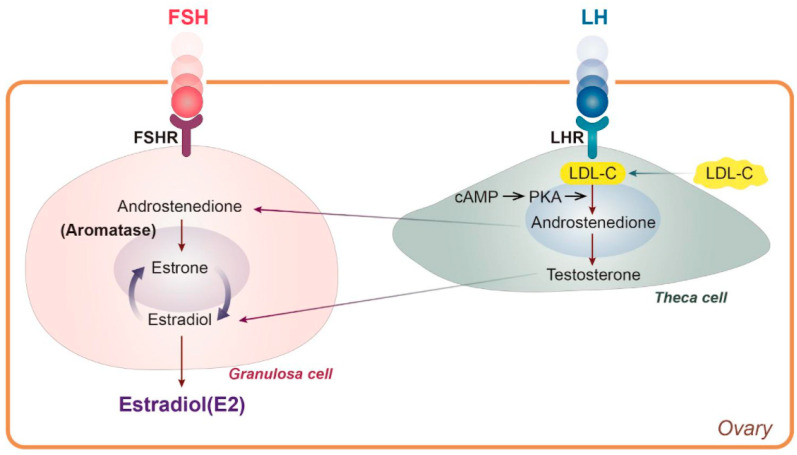
LDL cholesterol is used for E2 synthesis. cAMP: cyclic adenosine monophosphate, E2: estradiol, FSH: follicle-stimulating hormone, FSHR: follicle-stimulating hormone receptor, LDL-C: low-density lipoprotein cholesterol, LH: luteinizing hormone, LHR: luteinizing hormone receptor, PKA, protein kinase A.

**Figure 2 nutrients-13-04556-f002:**
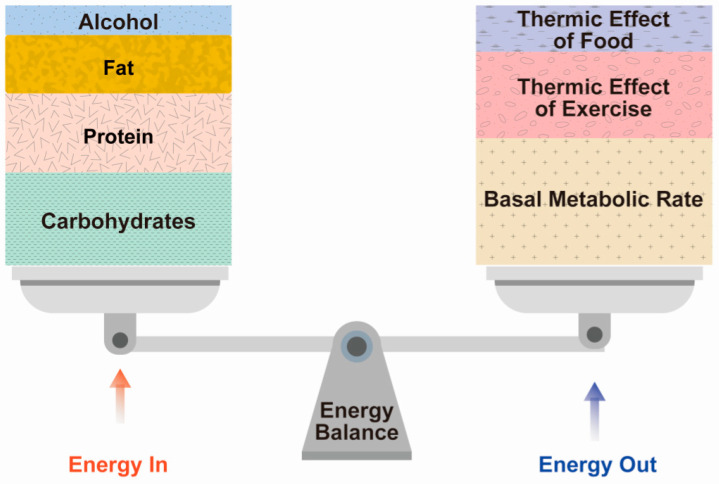
Energy balance between energy in and energy out.

**Table 1 nutrients-13-04556-t001:** Body compositions of Reference Men and Women [14].

	Men	Women
Age, years	20–24	20–24
Height, in	68.5	64.5
Weight, lb	154	125
Total fat, lb (% body weight)	23.1 (15.0%)	33.8 (27.0%)
Storage fat, lb (% body weight)	18.5 (12.0%)	18.8 (15.0%)
Essential fat, lb (% body weight)	4.6 (3.0%)	15.0 (12.0%)
Muscle, lb (% body weight)	69 (44.8%)	45 (36.0%)
Bone, lb (% body weight)	21 (14.9%)	15 (12.0%)
Remainder, lb (% body weight)	38.9 (25.3%)	31.2 (25.3%)
Average body density	1.070 g/mL	1.040 g/mL

**Table 2 nutrients-13-04556-t002:** Estimated metabolic rates of tissues and percentage contribution to total metabolism [34].

	REE	MEN	WOMEN
	kcal/kg/Day	% Total REE	% Total REE
Liver	200	17	18
Brain	240	19	21
Heart	440	9	8
Kidneys	440	8	8
Skeletal muscle ^a^	13	24	20
Adipose tissue	4.5	4	7
Other ^b^	12	19	18
Total		100	100

REE, resting energy expenditure. ^a^ Resting and nonexercised recovery rate; ^b^ skeleton, blood, skin, gastrointestinal tract, lungs, spleen, and other organs.

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
