# Peer review of "Energy Metabolism Changes and Dysregulated Lipid Metabolism in Postmenopausal Women"

_nutrients, 2021, doi:10.3390/nu13124556_

Round 1
Reviewer 1 Report
The publication by See-Hee Ko and YunJae Jung compiles an interesting informations. During the postmenopausal period of life, there are unfavorable changes in the women metabolism and that fact implement for several areas.
There are a few comments that I would like to mention:
Line 44-46: For better understanding, It should be separated for two sentences with at least two independent references. Cannot base everything on the one reference [8]. Check, for example M.L. Maltais et al.: Changes in muscle mass and strength after menopause.
Line 50-54: Authors pointed a lot of cross-sectional factors and issues. However, I believe it would be important to mention clearly in this part - if the reader should take this review as (for example) part of prophylaxis ??
Line 74-75: word “fuel” is unproper, may be - energy?
Line 195-208: Nowadays, a more appropriate and comprehensive presentation of these issues is the concept of Osteosarcopenia. Where we connect osteopenia/osteoporosis (low bone mass) with sarcopenia (low muscle mass). I believe that this issues should be presented in line. Check Kirk et al Osteosarcopenia: A case of geroscience
Moreover, This section should be completed with information about areas of fractures, mostly femoral neck due to result in significant morbidity and mortality.
Line 279-280: Authors pointed that: Extensive studies have been conducted on the role of estrogen in various metabolisms, immunity, and inflammatory processes in rodents and humans. At the end of this section it should be clearly marked how the results from one group influent on another. Here, readers may lose the limit of interpretation of results or indicate the most valuable results for humans.
Line 332-334: Authors pointed the advantage of review but it’s difficult to find the “take-home message”. What is this review for? Do you believe that your study makes a significant contribution to the literature? Point it more clearly.
In general, my biggest concern about this paper is lack of analysis of physical activity. However, authors mention it as limitation in conclusion. For future research, PA of various volumes is necessary to analyze.
Thank You.
Author Response
Referee #1
- Line 44-46: For better understanding, It should be separated for two sentences with at least two independent references. Cannot base everything on the one reference [8]. Check, for example M.L. Maltais et al.: Changes in muscle mass and strength after menopause.
We appreciate the reviewer’s comment. We have separated the sentence in the revised manuscript (line 44–48), as follows:
With these hormonal changes, menopausal women are more likely to experience various metabolic disorders such as dysregulated lipid metabolism, fat redistribution, visceral fat accumulation, and altered fatty acid metabolism [8-10]. Furthermore, it is readily accompanied by changes in body composition and energy metabolism, loss of muscle volume and strength, and weight gain [11-13].
- Line 50-54: Authors pointed a lot of cross-sectional factors and issues. However, I believe it would be important to mention clearly in this part - if the reader should take this review as (for example) part of prophylaxis.
Based on this suggestion, we have corrected the sentence in the revised manuscript (line 51–57), as follows):
This review describes the concepts needed to understand energy metabolism, including the composition of the human body and the components of energy expenditure, and discusses the possible changes in energy metabolism in postmenopausal women. By summarizing the overall energy metabolism, this review provides information to prevent muscle and bone loss. In addition, it also addresses lipid metabolic changes related to visceral fat accumulation, lipolysis, fatty acid oxidation, fat redistribution, and weight gain.
- Line 74-75: word “fuel” is unproper, may be - energy?
Following this suggestion, we have corrected “a fuel source” to “an energy source” in the revised manuscript (line 78).
- Line 195-208: Nowadays, a more appropriate and comprehensive presentation of these issues is the concept of Osteosarcopenia. Where we connect osteopenia/osteoporosis (low bone mass) with sarcopenia (low muscle mass). I believe that this issues should be presented in line. Check Kirk et al Osteosarcopenia: A case of geroscience. Moreover, this section should be completed with information about areas of fractures, mostly femoral neck due to result in significant morbidity and mortality.
We appreciate the reviewer’s comment. We have added corresponding descriptions of osteosarcopenia in the revised manuscript (line 205-222), as follows:
When sarcopenia coexists with osteoporosis, it results in a geriatric syndrome called “osteosarcopenia,” which increases the risk of weakness, hospitalization, and death [54]. In a prospective cross-sectional study by Buliana et al., the prevalence of osteosarcopenia was reported to be high among postmenopausal women with an increased risk of fracture [55]. Patients in the osteosarcopenia group had a greater risk of frailty than patients in the osteoporosis-alone group (odds ratio, 2.33; 95% confidence interval, 1.13–4.80; and P = 0.028). According to a retrospective observational study, one-year mortality of osteosarcopenia (15.1%) was higher than that of other groups (normal, 7.8%; osteoporosis only, 5.1%; and sarcopenia only, 10.3%) [56]. In postmenopausal women, insufficient protein and calcium intake and low levels of physical activity appear to be the most common risk factors for osteosarcopenia [55]. Generally, it is accepted that menopause is associated with accelerated loss of FFM or skeletal muscle, which further decreases energy expenditure during rest and physical activity [57]. Thus, body function impairment originates from muscle loss in menopausal women, leading to difficulties in carrying out voluntary activities and reducing the quality of life. The most promising strategy for increasing muscle and bone mass is resistance training, as well as sufficient amounts of protein, vitamin D, calcium, and creatine to help preserve these tissues during menopause [53].
- Line 279-280: Authors pointed that:Extensive studies have been conducted on the role of estrogen in various metabolisms, immunity, and inflammatory processes in rodents and humans. At the end of this section it should be clearly marked how the results from one group influent on another. Here, readers may lose the limit of interpretation of results or indicate the most valuable results for humans.
Following these suggestions, we have added a corresponding description in the revised manuscript (line 309-315), as follows:
Estrogens are also associated with inflammatory responses in women. Although estrogens are known to enhance autoimmune diseases [81], decreased estrogen increases susceptibility to infectious diseases, as demonstrated by the defective innate immune responses against viral infection in OVX mice [82]. In addition, OVX enhanced the susceptibility of female rats to dyslipidemia with a decrease in innate cytokines, suggesting impaired metabolic and immune homeostatic responses with the loss of estrogens [83].
- Line 332-334: Authors pointed the advantage of review but it’s difficult to find the “take-home message”. What is this review for? Do you believe that your study makes a significant contribution to the literature? Point it more clearly.
We appreciate this comment. We have stated the “take-home message” of the manuscript (line 349-355), as follows:
The advantages of this review are that energy metabolism and lipid metabolic disorders related to menopause are comprehensively summarized in the basic concepts of body composition and energy expenditure. By understanding the overall energy metabolism, it will be possible to address individual health by understanding osteosarcopenia and recognizing the importance of preventing muscle loss. This review also systematically organized the background of weight gain due to E2 loss and lipid metabolic abnormalities.

Reviewer 2 Report
This is a well written review article detailing metabolic, especially lipid, changes with age in women. It is well-referenced with descriptive figures. The authors have done a good job of compiling such a valuable resource for their colleagues. I would advise the authors to go over the the manuscript and make corrections to grammar, syntax, and spellings before final submission.
Author Response
Referee #2
- This is a well written review article detailing metabolic, especially lipid, changes with age in women. It is well-referenced with descriptive figures. The authors have done a good job of compiling such a valuable resource for their colleagues. I would advise the authors to go over the manuscript and make corrections to grammar, syntax, and spellings before final submission.
We appreciate this comment. We have checked the use of English and corrected the errors in the revised manuscript.
We deeply appreciate the valuable suggestions provided by the reviewer and hope that the current version of the manuscript is suitable for publication.